# Identification of Ballast Fouling Status and Mechanized Cleaning Efficiency Using FDTD Method

Bo Li [1], Zhan Peng [2], Shilei Wang [2] and Linyan Guo [1,*]

1    School of Geophysics and Information Technology, China University of Geosciences, Beijing 100083, China; lib@email.cugb.edu.cn
2    Infrastructure Inspection Research Institute, China Academy of Railway Sciences Corporation Limited, Beijing 100081, China; pengzhan@rails.cn (Z.P.); wangshilei@rails.cn (S.W.)
*    Correspondence: guoly@cugb.edu.cn

**Abstract:** Systematic assessment of ballast fouling and mechanized cleaning efficiency through ground penetrating radar (GPR) is vital to ensure track stability and safe train transportation. Nevertheless, conventional methods of ballast fouling inspection and evaluation impede construction progress and escalate the cost of maintenance. This paper proposes a novel method using random irregular polygons and collision detection algorithms to model the ballast layer and simulated using the finite-difference time-domain (FDTD) algorithm. Hilbert transform energy, S-transform, and energy integration curve are employed to identify ballast fouling and cleaning efficiency. The highly fouled ballast exhibits concentrated Hilbert transform energy, increased energy attenuation rate in S-transform with depth in the 1.0–3.0 GHz, along with a stronger energy integration curve. Clean or post-cleaning ballast shows opposite results. Experiments on a passenger trunk line in southern China validated the method's accuracy after mechanized ballast cleaning. This approach guides GPR-based detection and supports railway maintenance. Future studies will consider heterogeneous properties and the three-dimensional structure of the ballast layer.

**Keywords:** ballast fouling status; ballast mechanized cleaning; FDTD; Hilbert energy transform; S-transform; integral energy curves; GPR





## 1. Introduction

The ballast layer plays a vital role in providing elastic support and drainage for train tracks. However, it is subjected to mechanical wear, deterioration, and fouling, posing a significant risk to track stability and safety [1]. Frequent assessments of the ballast condition are necessary for minimizing maintenance costs and preventing accidents, which is critical for railroad operation and maintenance. While conventional methods adhere to international standards and yield reliable information, they are limited to discrete data and are labor-intensive [2]. Therefore, non-destructive testing techniques have the potential to efficiently and continuously acquire data on the fouling of railway ballast.

Ground penetrating radar (GPR) is a non-destructive technology that is increasingly used for condition assessment and safety monitoring of railway ballast fouling. This method relies on electromagnetic wave theory and has important applications in different fields, such as civil and environmental engineering [3], archaeology [4], and planetary exploration [5,6]. Early researchers in railway engineering used low-frequency GPR for testing [7], but were unable to interpret most of the results due to the low resolution. Subsequent researchers increased the frequency of GPR used for better resolution [8]. In addition, models for assessing the condition of railroad ballast and predicting the fouling of the ballast based on GPR are available in references [9–11]. GPR models for railroad ballast are costly and challenging due to the complex nature of the ballast and the variability of physical conditions. These procedures require effort and time, and they may result

in erroneous results. Numerical simulations are able to model the physical systems on a computer, providing valuable insights for researchers and assisting in design optimization.

In recent years, A. Benedetto et al. have proposed a GPR-based approach for assessing the cleanliness and fouling conditions of railway ballast layers using the random sequential adsorption (RSA) algorithm and finite-difference time-domain (FDTD) methodology [12]. Another research by L. Bianchini Ciampoli et al. examined the effects of particle size on the propagation and scattering of electromagnetic waves in railway ballast layers using the gprMax 2D numerical analysis software [13]. Additionally, the FDTD algorithm has been employed in the study of asphalt concrete, where the modeling of circle-shaped aggregates has been used to assess the condition of the pavement containing water [14–16]. However, since the numerical simulation method is limited to generating only spherical particles, the simulation results may not accurately reflect real-world scenarios and should be considered as a reference for further studies. Building on these studies, references [17,18] employed a discrete element method to generate three-dimensional irregular polygonal particles for simulating Martian rock formations, which were used to predict and guide exploration of subsurface water on Mars using fully polarized GPR. Nevertheless, the limited quantity of generated irregular three-dimensional particles render them unsuitable for modeling railway ballast.

Inspired by such research, the main contributions of this paper are as follows,

- This paper proposes a novel algorithm for generating random irregular polygon (RIP) particles to simulate railway ballast particles, addressing the issue of uniform and unrealistic particle shapes. In addition, an efficient algorithm is presented to generate the ballast layer by collision detection (CD) of a large number of particles, and then the particle size distribution is controlled to simulate the gradation of the ballast. The generated geological models can represent different levels of ballast fouling and the cleaning efficiency of ballast.
- Using the FDTD algorithm in forward simulation, the numerical simulation results accurately determine and identify the differences between the ballast layers with different ballast fouling and the efficiency of mechanized ballast cleaning process by integrating the energy curve, Hilbert transform energy [19–21], and S-transform time-frequency analysis [22,23].
- Finally, experiments were conducted on a section of a high-speed railway line in southern China with screened ballast. By comparing the results of the GPR forward simulation and the experimental data, we found that the simulation results were consistent with the measurements, indicating the accuracy and reliability of the proposed model.
- The remainder of this paper is organized as follows. Section 2 presents the materials, basic algorithms, and principles of data analysis used in this study. Section 3 provides an analysis of the simulation results for different levels of fouling and before and after ballast cleaning models. Section 4 describes the experimental content designed to compare the simulation results presented in Section 3. Finally, Section 5 summarizes the main contributions of this study.

## 2. Materials and Methods

### 2.1. Materials Preparation and Characterization

This study selected a 50 km downlink section of the intercity railway trunk line between two provinces in southern China for testing. Among them, 23 km was mechanized ballast cleaning in March 2023, while the rest were not performed, as shown in Figure 1a. The railway centerline was selected as the measurement line, with a track gauge of 8.3 cm and a time window of 15 ns. The radar equipment used a GSSI-SIR30 four-channel mainframe and a 4200S 2 GHz air-coupled antenna. Before the test, three-point sleeper box areas were selected in the before and after ballast mechanized cleaning sections, respectively, to excavate samples for grading analysis of the ballast, as shown in Figure 1b for excavation site photos and Figure 1c for particle size distribution after sieving. The relative dielectric

constant of the ballast layer was measured to be 6, while that of the sand cushion layer and the subgrade layer was measured to be 12 and 15, respectively, through experimental testing. Specific test methods can be found in Appendix A.

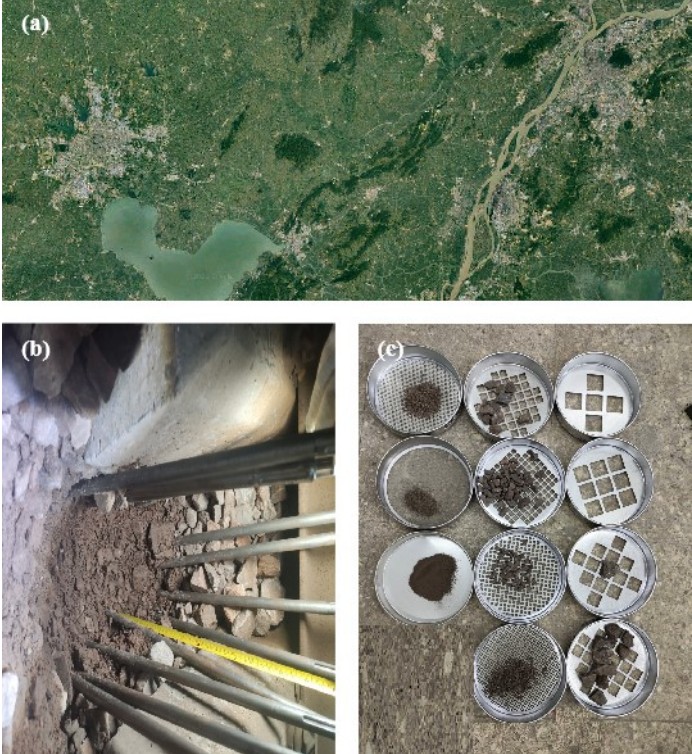

**Figure 1.** Experimental site (**a**) Google satellite map of the region between two provinces in southern China, (**b**) excavation site photo of the railway ballast bed, (**c**) sieved track ballast.

*2.2. 2D RIP Ballast Modeling Algorithm*

In this paper, we present a simple and efficient algorithm for establishing, placing, and overlapping a two-dimensional RIP ballast model. All location and shape information for the ballast is stored, called, and evaluated in the matrix format. Compared to traditional loop-based methods, this algorithm is much more efficient and faster in generating a large number of particles.

The two-dimensional RIP ballast can be abstracted and equivalent to a RIP shape. This equivalence is more in line with the real ballast model than the traditional approach of directly replacing circles. The RIP ballast is built based on circles by dividing them equally using the number of polygon sides and randomly increasing or decreasing the length on their radii. Figure 2 shows the model of RIP ballast generated. The main determining factors of RIP ballast particles are four, namely, the radius $r$ of the basic circle, the number of sides $k$ of the irregular polygon, the irregular shape control variable $\Delta r$, and the arc angle $\Delta \theta$ corresponds to each side.

Based on this, a flow chart for generating RIP particles of ballast is shown in Figure 3a. Firstly, the average particle size $r_{avg}$ and the number of sides $k$ of the particle need to be determined. Using $k$ as the number of iterations in the loop, the circle with a radius of $r_{avg}$ is divided into $k$ parts. Thus, the included angle after division can be obtained as $\theta_m = \frac{2\pi m}{k}$, where $m = 1, 2, 3, \ldots$. Finally, the length $\Delta r$ randomly changed based on the circle is given for each vertex of the randomly generated $k$-sided polygon. As a result, the position coordinates of each vertex (1) can be obtained. All positions will be saved in the form of matrices, which facilitates the CD of the subsequently deposited ballast.

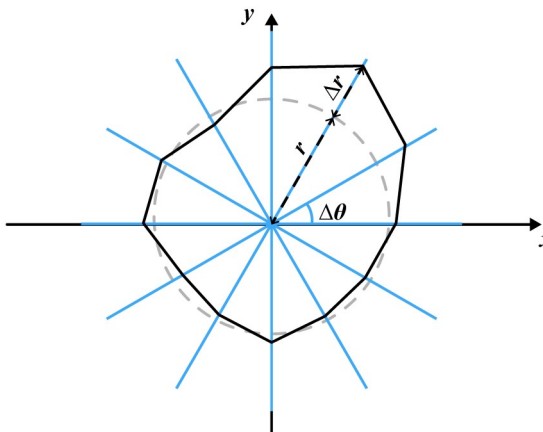

**Figure 2.** RIP Ballast Particle Model.

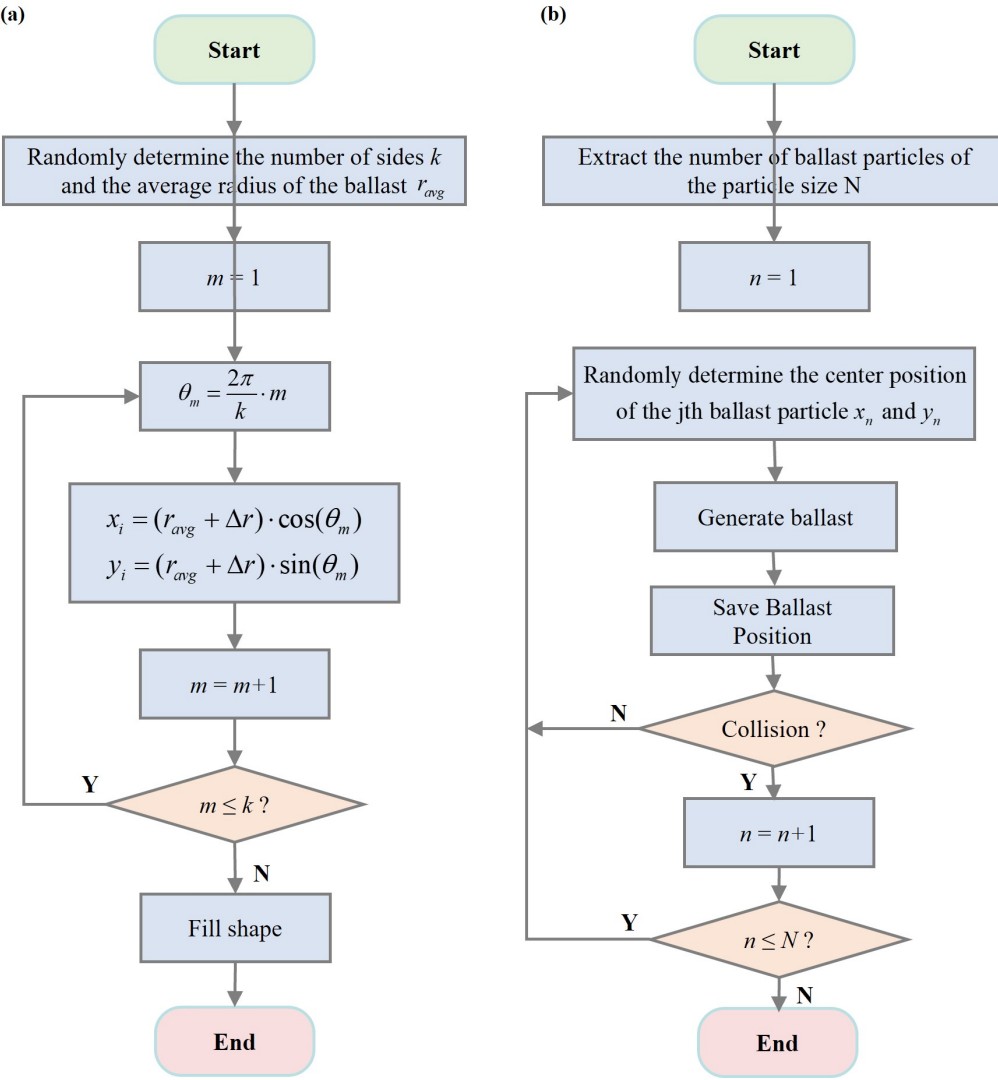

**Figure 3.** (**a**) Algorithm flowchart for generating RIP ballast particles, (**b**) Algorithm flowchart for ballast particles placement and CD.

After the generation of the RIP ballast particles, the next step is to distribute ballasts of different sizes into the designated area. This process mainly involves the CD between ballast particles and the speed of particle distribution. To this end, this study derives the

conditions for determining the collision of ballast particles and provides conditions for non-collision using a matrix approach. Moreover, this study addresses limitations related to the use of loops, such as low efficiency and high complexity in CD, which are not conducive to the generation of a large number of particles. The algorithm flowchart for the placement and CD of roadbed ballast particles is presented in Figure 3b. Referring to the schematic diagram shown in Figure 1, the matrices of the $X$, $Y$, and $R$ coordinates of the centers and radii of $n - 1$ RIP for the ballast particles can be represented by Equation (2).

$$\begin{aligned} x_i &= \left(r_{\text{avg}} + \Delta r\right) \cdot \cos(\theta_m) \\ y_i &= \left(r_{\text{avg}} + \Delta r\right) \cdot \sin(\theta_m) \end{aligned} \tag{1}$$

$$\begin{aligned} X &= [x_1, x_2, x_3, \ldots, x_{n-1}] \\ Y &= [y_1, y_2, y_3, \ldots, y_{n-1}] \\ R &= [r_1, r_2, r_3, \ldots, r_{n-1}] \end{aligned} \tag{2}$$

$n = 1, 2, 3, \ldots$ represents the $k$th generated particle. Subsequently, the distance matrix $D$ between the $n$th and the $(n-1)$th particles can be calculated using Equation (3).

$$\begin{aligned} D &= \sqrt{(x_n - X)^2 + (y_n - Y)^2} \\ &= [d_1, d_2, d_3, \ldots, d_{n-1}] \end{aligned} \tag{3}$$

The distance $\Delta D$ between the road stone particles can be expressed as Equation (4).

$$\begin{aligned} \Delta D &= D - (R + r_k) \\ &= [d_1 - r_1 - r_k, d_2 - r_2 - r_k, \ldots, d_{k-1} - r_{k-1} - r_k] \end{aligned} \tag{4}$$

Therefore, it is not difficult to derive the condition for determining that there is no collision between the particles, which is expressed as $min(\Delta D) \geq 0$. During the implementation of the algorithm, if a collision occurs among the particles, the process will return to the step of selecting the particle positions, until all the ballast particles are generated without collision. In the CD of the ballast particles, it is only necessary to find a minimum value that satisfies the condition to avoid collision. Compared with repeatedly judging the distance difference through loops to determine the collision, this approach significantly improves the generation speed and efficiency, particularly when the number of ballast particles is large.

### 2.3. Hilbert Transform Energy

In the data analysis of this paper, the Hilbert transform is utilized to extract the envelope of the reflection pulse signal, which is used to characterize the power of the reflection signal. The Hilbert transform of the signal $s(t)$ can be represented as the convolution of $s(t)$ and $h(t) = \frac{1}{\pi t}$ [19,20], and can be expressed in Equation (5).

$$\hat{s}(t) = h(t)^*s(t) = \int_{-\infty}^{\infty} \frac{s(\tau)}{t - \tau} d\tau \tag{5}$$

The application of Hilbert transform to the GPR echo signal $s(t)$ yields Equation (6).

$$s_a(t) = s(t) + i\hat{s}(t) \tag{6}$$

where $\hat{s}(t)$ represents the Hilbert transform of the input signal $s(t)$. Consequently, the instantaneous amplitude (envelope amplitude) of the GPR echo signal $s(t)$ can be obtained as Equation (7).

$$|s_a(t)| = \sqrt{s(t)^2 + \hat{s}(t)^2} \tag{7}$$

### 2.4. Time-Frequency Analysis of S-Transform

The time-frequency analysis has significant advantages in recognizing the level of the ballast fouling, as it is more capable of distinguishing the clean ballast layer from the transition zone to the highly fouling ballast than commonly used threshold estimation techniques, such as spectral analysis. The common time-frequency analysis methods for processing GPR signals include short-time Fourier transform and S-transform. Short-time Fourier transform is limited by the choice of window function, whereas S-transform overcomes this limitation [24,25]. For a single trace, a large number of traces are required to obtain a time-frequency radar image. To obtain a time-frequency representation, the S-transform uses functions based on continuous wavelet transform as shown in Equation (8).

$$W(\tau, d) = \int_{-\infty}^{\infty} h(t) \cdot w(t - \tau, d)dt \tag{8}$$

where $h(t)$ represents a given function, $d$ is a dilation factor that controls the time-frequency resolution trade-off, and $w(t, \tau)$ denotes the scaling replica of the mother wavelet used for decomposition. Therefore, the time-frequency representation of the given trace $h(t)$ using the S-transform is defined by multiplication with a phase factor in both time and frequency domains.

$$S(t, f) = e^{i2\pi f\tau} \cdot W(\tau, d) \tag{9}$$

## 3. Modeling and Simulation Result Analysis

### 3.1. Modeling and Simulation Analysis of Ballast Fouling

To delineate the particle size distribution of ballast particles for different degrees of the ballast fouling, this paper sieved the ballast particles from clean, moderately fouled, and highly fouled ballast excavated on-site, and obtained their corresponding particle size distributions as shown in Table 1. It is evident from Table 1 that the essential difference between the clean and highly fouled ballast is in the sieving rate of the 1–35 mm particles. It is substantially higher for the highly fouled ballast, while clean ballast particles are mostly distributed between 35–63 mm in diameter. This suggests that the finer ballast particles tend to accumulate on highly fouled ballast or older ballast layer sections, which is beneficial for modeling the fouling ballast at different levels in subsequent analyses.

**Table 1.** Gradation table of ballast particles.

| Particle Size | Fouling Level | | |
|---|---|---|---|
| | Clean | Moderately Fouled | Highly Fouled |
| 1.0–2.5 mm | 2.2% | 3.8% | 5.3% |
| 2.5–5.5 mm | 3.1% | 4.9% | 7.0% |
| 5.0–10.0 mm | 5.2% | 7.3% | 9.8% |
| 10.0–16.0 mm | 9.3% | 13.7% | 16.9% |
| 16.0–25.0 mm | 14.9% | 20.0% | 26.6% |
| 25.0–35.0 mm | 32.0% | 34.7% | 41.1% |
| 35.0–45.0 mm | 54.2% | 55.8% | 58.3% |
| 45.0–56.0 mm | 79.0% | 81.3% | 83.1% |
| 56.0–63.0 mm | 95.1% | 96.0% | 97.5% |

Based on the aforementioned ballast particles size results, this study uses Python 3.10 under PyCharm and Anaconda to model clean, moderately fouled, and highly fouled ballast and conducted electromagnetic simulations using the open-source software gprMax based on the FDTD technique [26]. Randomly sized and RIP ballast particles are introduced in the model to generate the ballast layer with different particle size distributions, in accordance with the relationship and conclusions between particle size and levels of fouling are defined in Table 2. Figure 4a–c shows three different cases of ballast fouling, visualized using the software named Paraview with vti files, with a length and height of

5.0 m and 2.5 m, respectively. The mesh parameters of gprMax were set to $dx\_dy\_dz = [0.001, 0.001, 0.01]$. In the FDTD simulation, a dipole antenna built into the software was chosen as the transmitting and receiving antennas, with a distance of 0.05 m from the railway and a height of 0.3 m. The antenna was scanned in the lateral direction of $x$ with a step size of 0.05 m. To accelerate the simulation in gprMax, a 6 GB RTX3060 GPU was utilized. In order to avoid the influence of the absorbing boundary of gprMax on the electromagnetic waves of the antennas, 80 AScan results were obtained by scanning the antennas along 80 traces between 0.5 m to 4.5 m in the x-axis direction with a step size of 0.05 m.

**Table 2.** Electromagnetic parameters of gprMax models at different levels [27].

| Model Level | Dielectric Constant | Permeability (H/m) | Conductivity (S/m) |
|---|---|---|---|
| Air layer | 1 | 1 | 1 |
| Ballast layer | 6 | 1 | 0.01 |
| Sand cushion layer | 12 | 1 | 0.001 |
| Subgrade layer | 15 | 1 | 0.01 |

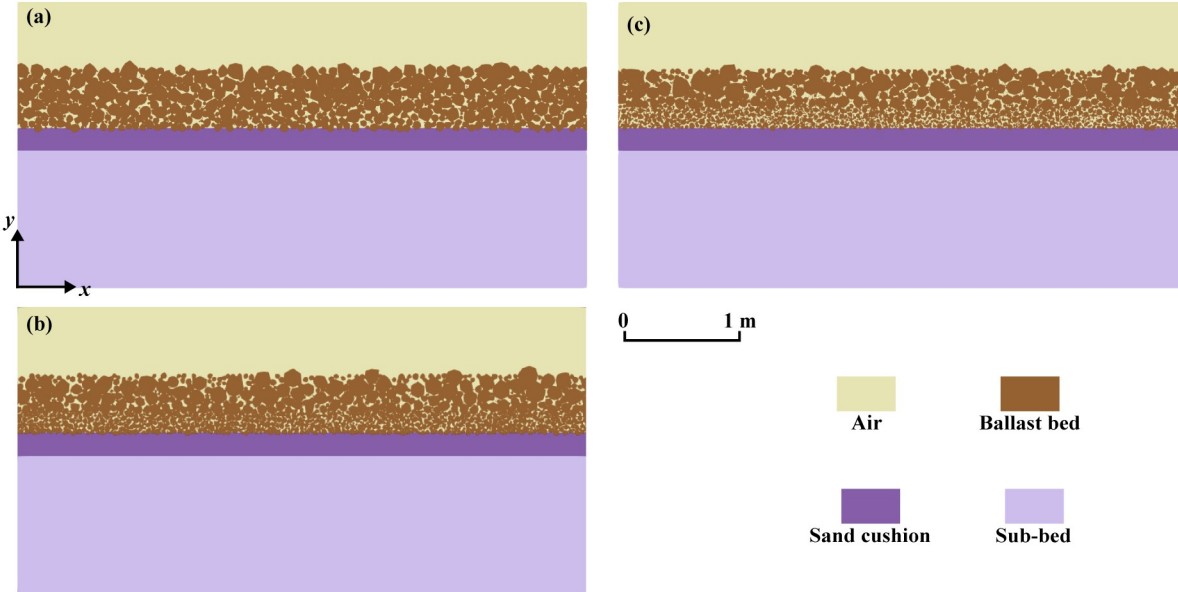

**Figure 4.** Different levels of gprMax modeling and visualization for different degrees of contamination: (**a**) clean ballast bed, (**b**) ballast bed with moderate contamination, (**c**) ballast bed with severe contamination.

As shown in Figure 5, the BScan, Hilbert transform energy, and S-transform time-frequency results obtained from the simulations correspond to the three different levels of the ballast fouling, respectively. Figure 5a–c, respectively, depict the BScan simulation results for the three different levels of the ballast fouling, with a linear gain of up to 20 applied to better observe the stratification effect of the medium, without removing the direct wave. The more detailed data processing methods are shown in Appendix B. A clear boundary between the ballast layer and the sand cushion layer can be observed at a time window of approximately 8.8 ns. In addition, as the level of fouling increases, the total BScan reflections become more cluttered and the reflected electromagnetic waves become stronger. Furthermore, in order to further distinguish between different levels of ballast fouling, this paper processed the data with a Hilbert transform energy, in which the background direct wave was first removed before the transformation, without linear gain applied, to avoid affecting the transformation results.

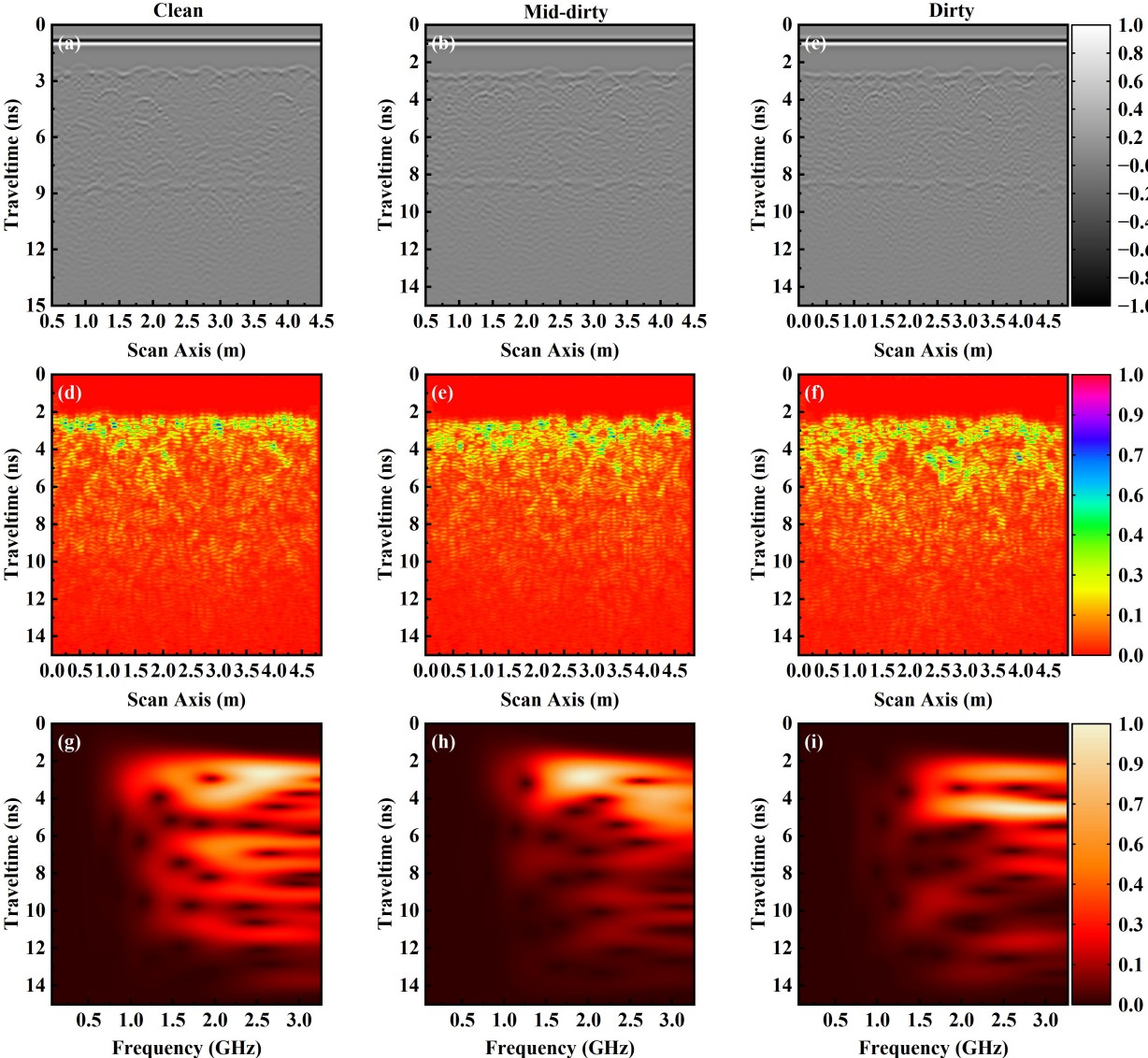

**Figure 5.** Forward simulation results of clean, moderately fouled, and highly fouled ballast: (**a**–**c**) show the BScan grayscale images; (**d**–**f**) depict the Hilbert transform energy results; and (**g**–**i**) demonstrate the S-transform time-frequency results at a depth of 2.5 m.

Figure 5d–f shows the Hilbert transform energy results for different levels of fouling. To facilitate comparison, the energy was normalized. It can be observed from the figures that as the fouling increases, the energy distribution in deeper layers becomes denser, indicating stronger energy in the small-sized ballast layer. To further observe the difference between different fouling and the fouling transition zone, this study used fixed single-track simulated data to obtain time-frequency energy results through S-transform, as shown in Figure 5g–i. Firstly, the energy was normalized using the maximum value, and then the simulated data were taken at 2.5 m to obtain the time-frequency results through S-transform. By comparing the figures, it can be found that the S-transform energy for different ballast fouling is mainly concentrated in the range of 1.0 GHz to 3.0 GHz. Furthermore, by comparing the rate of energy decay, it can be observed that with increasing fouling, the decay rate of electromagnetic energy corresponding to S-transform increases as the longitudinal depth extends. For the clean ballast, the high-frequency energy remains high until about 12 ns, while for the moderately fouled ballast, the high-frequency energy decays completely at around 6 ns. In contrast, for highly fouled ballast, significant decay starts at around 5 ns. Therefore, S-transform is more effective in distinguishing different

degrees of fouling in the ballast, and the decay rate of electromagnetic energy corresponding to S-transform increases with increasing fouling degree.

### 3.2. Model and Simulation Analysis of Mechanized Ballast Cleaning on Railway Ballast

This study established the model of mechanized ballast cleaning process efficiency, as illustrated in Figure 6. The model has a total length of 15 m and a height of 2.5 m. The first and last 5 m of the model were designed to represent highly fouled ballast, representing the condition without mechanized ballast cleaning. The model's center displays clean ballast that is up to 5 m long and represents the circumstances in a ballast cleaning. Simulation is performed using gprMax, with parameters such as antenna and spacing between tracks kept constant. The antenna acquired 260 traces with a step size of 0.05 m from 1 m to 14 m. The final simulation results are presented in Figure 7. In order to better highlight the contrasting effect of energy changes that occurred by mechanized ballast cleaning efficiency, this study calculated and obtained the integrated energy of each trace line, plotted the energy curve, and smoothed and normalized the energy values to a range of 0–1, as shown in Figure 7a. It can be observed that the normalized energy of highly fouling areas is around 0.8, while the normalized energy of the clean ballast with ballast cleaning is very close to 0, indicating that the energy of the ballast after mechanized cleaning is much lower than that of it before ballast cleaning. This conclusion is consistent with the results of the previous section that the energy of the clean ballast is lower than that of the highly fouled ballast. By observing Figure 7b,c, it is easy to find that the energy distribution of the clean ballast layer after mechanized ballast cleaning becomes sparser as the depth increases, while the energy distribution of the before ballast cleaning on both sides becomes denser as the depth increases.

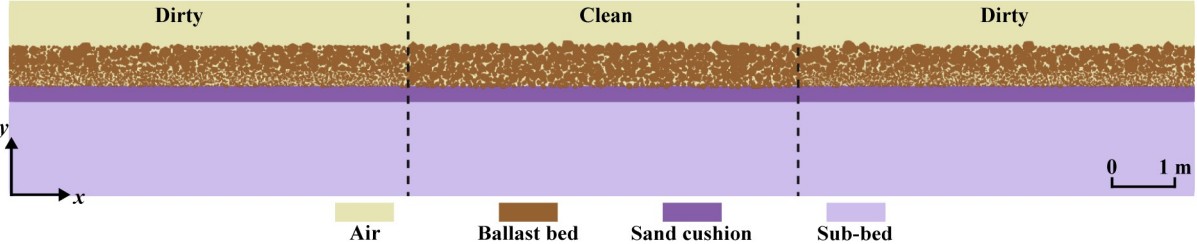

**Figure 6.** The gprMax model before and after ballast cleaning.

Furthermore, this study obtained the corresponding time-frequency results of the S-transform for three different points at 2.95 m, 7.1 m, and 13.2 m, respectively, from three different ballast layers sections before, after, and before mechanized ballast cleaning, and normalized the results using the maximum value. By comparing the results, it can be analyzed from the energy decay rate that the 1–6 GHz energy of the after ballast cleaning case generally decays more slowly, while the energy decay rate of the before ballast cleaning case is extremely fast, which is helpful to distinguish and judge whether the railway section has been subjected to an efficient mechanized ballast cleaning process.

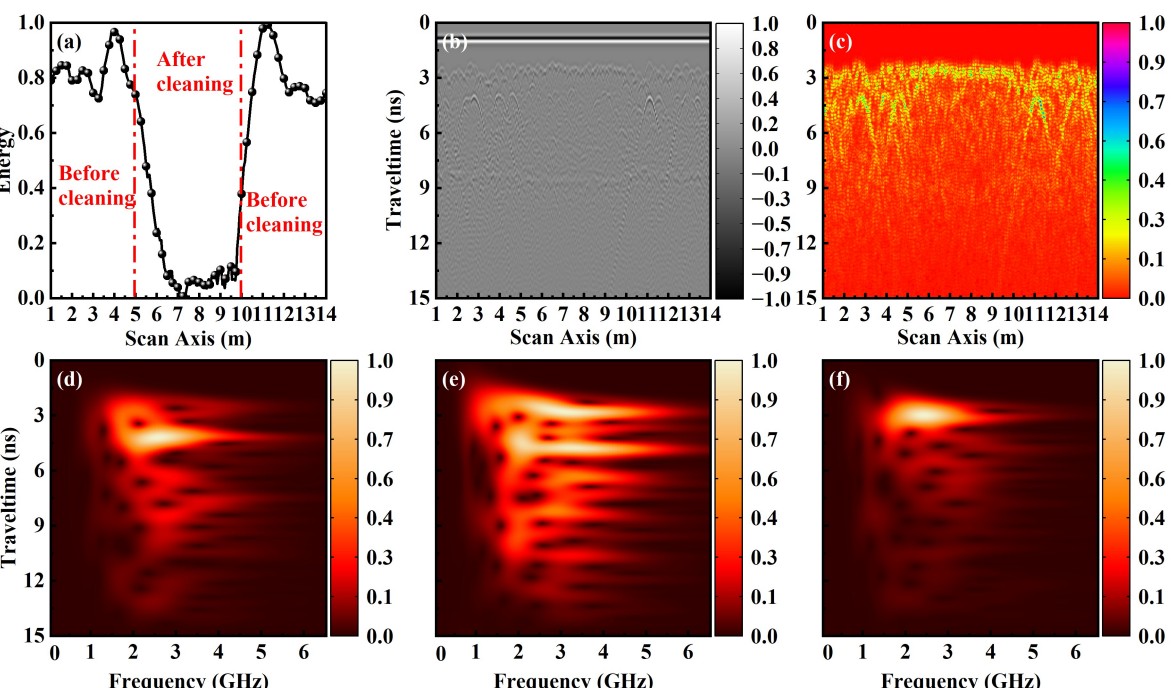

**Figure 7.** Different simulation results before and after ballast cleaning, with (**a**) energy change curve, (**b**) BScan grayscale image, (**c**) Hilbert transform energy map, and (**d**–**f**) S-transform energy results at 2.95 m, 7.1 m, and 13.2 m, respectively.

## 4. Experimental and Analysis

To compare and validate the consistency between the simulation and experimental results, selected sections of the main railway line were measured in this study, as shown in Figure 8a. To address the abnormal data caused by environmental noise at the rail joints, we directly screened and removed them based on the abnormality of the integrated energy, as shown in Figure 8b for the integrated energy curve of the entire railway section. After normalization, it was found that the energy clearly decreased in the 25–42.5 km section, with a minimum value of around 0, consistent with the results for the selected section of the mechanized ballast cleaning location, while other sections were uncleaned and maintained an energy level of around 0.8. This provides evidence that although the scale of the simulation method proposed in this study is limited to 1–14 m, the energy trends of the entire section before and after mechanized ballast cleaning remain consistent. In addition, the integrated energy curve indicates that cleaner ballast has lower energy levels, whereas more severe levels of ballast fouling correspond to higher energy levels.

To further analyze this experimental result, this paper extracted data from 0–60 tracks and 300,000–300,060 tracks, respectively, for analysis, corresponding to the results before and after ballast cleaning or clean ballast and highly fouled ballast. For this part of the data, the direct waves of the experimental data were first zero-corrected and removed, and the repeated sleeper information in the experiment was eliminated by truncation, so the results in Figure 9a,b,d,e are relatively flat at around 4 ns. As shown in Figure 9a–c, they, respectively, correspond to the BScan grayscale image, Hilbert transform energy image, and S-transform time-frequency image before ballast cleaning.

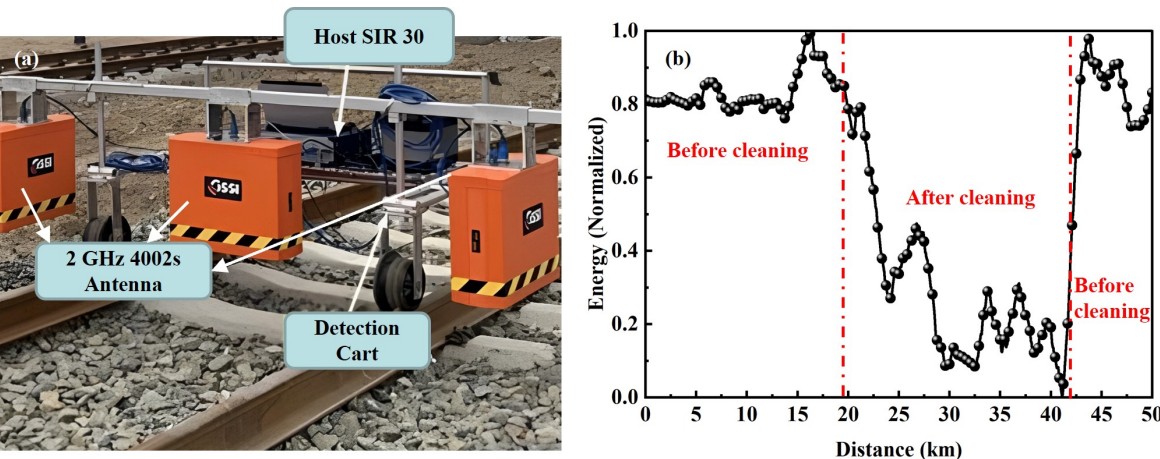

**Figure 8.** The (**a**) test scene and (**b**) integrated energy curve of a railway trunk line in southern China.

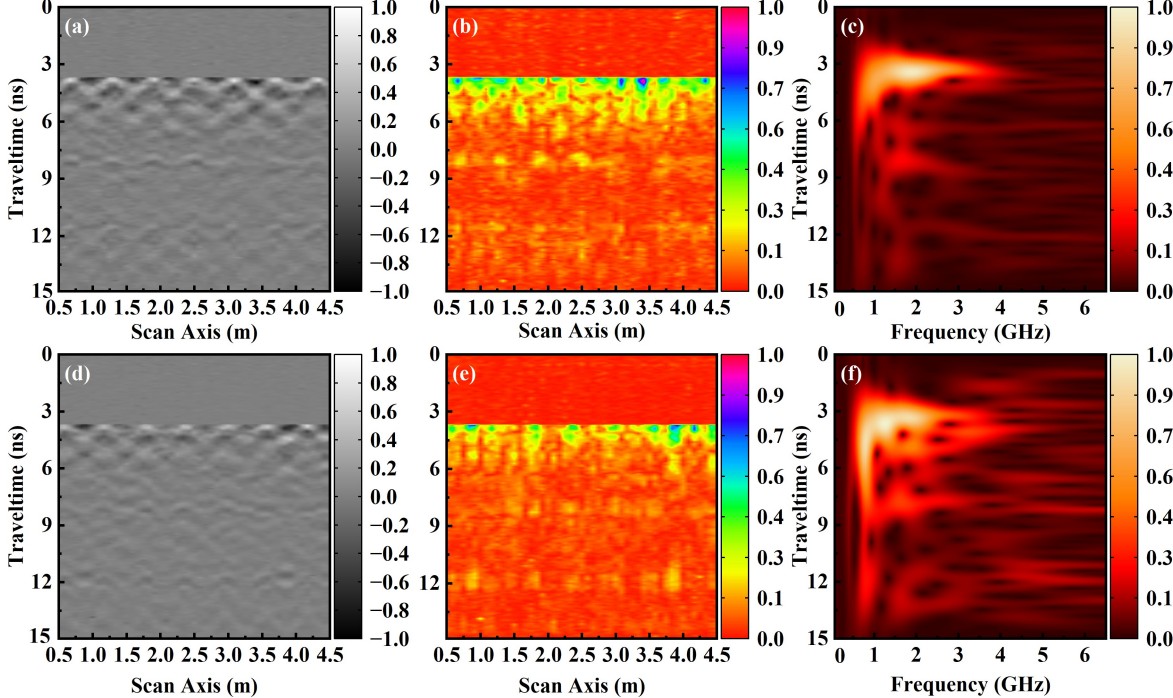

**Figure 9.** The experimental data before and after ballast cleaning are presented in (**a**,**d**) as BScan grayscale images, (**b**,**e**) as Hilbert energy transformation graphs, and (**c**,**f**) as time-frequency spectrograms of the 30th intermediate data.

Based on the experimental results, the present study further analyzed the measured data of 0–60 tracks and 300,000–300,060 tracks, corresponding to the results before and after the mechanized ballast cleaning process can be understood as the results of clean ballast and highly fouled ballast. Firstly, the direct waves in the experimental data are removed, and the duplicated sleeper information in the experiment was eliminated by truncation. The detailed data processing methods are shown in Appendix B. Thus, the results of Figure 9a,b,d,e were relatively flat at around 4 ns. As shown in Figure 9, the gray-scale image of the BScan, the Hilbert transform energy distribution map, and the S-transform time-frequency distribution map corresponded to the results before ballast cleaning. Figure 9a,d show a faint boundary at around 8 ns and 12 ns, respectively. Before ballast cleaning, fine particles were primarily found in the lower layer. This resulted in a dense energy distribution in the Hilbert transform shown in Figure 9b,e. However, after ballast cleaning, the energy distribution became relatively sparse, with the majority

concentrated in the surface layer of the ballast. In addition, during the processing of experimental data, this study also selected the 30th track data between two datasets for the time-frequency energy analysis of the S-transform. The energy was also mainly concentrated at around 1.0–6.0 GHz. As the longitudinal depth increases, the energy decay rate before ballast cleaning is faster than after ballast cleaning. At around 6 ns, the high-frequency energy basically decayed in the before-ballast cleaning data, while in the after-ballast cleaning data, the energy had not yet completely decayed at around 14 ns at 1–6 GHz. Therefore, the processing and analysis of experimental data can fully explain that the ballast model and methods proposed in this study with different ballast fouling and mechanized cleaning process can interpret well the actual ballast fouling, and guide the analysis and judgment of the subsurface GPR-based ballast fouling detection in practical engineering.

This study compared the proposed method with a few pertinent research techniques, as given in Table 3, to further highlight the novelty of the approach. Comparative analysis revealed that the proposed method's two-dimensional RIP particle shape is more in line with the actual railway ballast and subgrade scenario. Additionally, the CD algorithm using matrix computation is highly efficient in dealing with RIP ballast scenarios with large particle counts. Therefore, it has good adaptability in modeling the railway ballast and can accurately reproduce its geological model with high precision. Although the two-dimensional ballast layer model is straightforward and simple to replicate, there is still a discrepancy between it and the real three-dimensional model. Since every ballast particle in the simulation is similar to a homogeneous, isotropic medium, there is still some experimental room for improvement. The ballast bed model approach also disregards the impact of other elements, such as the amount of water in the ballast, on the level of fouling.

**Table 3.** Comparison of Reference Methods and Our Approach.

| Reference | Modeling Algorithm | Particle Shape | Number of Particles |
|-----------|--------------------|-----------------|---------------------|
| [12,13] | random sequential adsorption | circle | many |
| [14–16] | random sequential adsorption | circle | many |
| [16] | discrete element algorithm | Irregular 3D polygon | few |
| This paper | RIP & CD algorithm | Irregular 2D polygon | many |

## 5. Conclusions

This paper presents a physically based railway ballast model for different levels of fouling and mechanized ballast cleaning process efficiency, implemented through a novel Python algorithm using the gprMax simulation software.

- Based on the proposed method, this study employs the energy integration curve, Hilbert energy transform, and S-transform approaches to evaluate and provide a reasonable analysis of the simulated results on different ballast conditions. It can effectively distinguish the railway ballast with different levels of fouling and mechanized ballast cleaning process. The conclusions of the simulation analysis are consistent with experimental data from a high-speed railway line in southern China, demonstrating the reliability and accuracy of the proposed model.
- As the fouling increases, the finely fragmented particles in the ballast layer tend to be more abundant and the energy on the integration curve also increases. In the Hilbert transform energy, the energy distribution is more concentrated. In the S-transform time-frequency results, the attenuation rate is faster with increasing depth. Conversely, with the clean ballast, the opposite effect is observed.
- This method is proposed can accurately reproduce the railway ballast model with high precision. Furthermore, the proposed research method accurately constructs models of different levels of fouling and mechanized ballast cleaning efficiency of the railway ballast, which has great potential application value in the detection and elimination of hidden dangers on actual railway lines.

- Nevertheless, the ballast layer model proposed in this paper is still a two-dimensional structure compared to the actual three-dimensional model situation there are still some differences. The relative permittivity of the ballast is only considered as a homogeneous and isotropic medium for the simulation and more environmental effects on the fouling of the ballast bed have not been considered. To address these flaws, further studies will expand the model to a three-dimensional structure and take into account the variation of relative permittivity and some natural climatic effects such as rainfall.

Finally, the method proposed in this paper could provide more scientifically accurate guidance and explanation for the actual measurement of GPR data on railway lines.

**Author Contributions:** Conceptualization, B.L.; Methodology, L.G.; Software, B.L. and Z.P.; Validation, Z.P.; Formal analysis, B.L.; Investigation, B.L. and Z.P.; Writing—original draft, B.L.; Supervision, L.G.; Project administration, Z.P., S.W. and L.G.; Funding acquisition, Z.P., S.W. and L.G. All authors have read and agreed to the published version of the manuscript.

**Funding:** This research was funded by the Foundation Project of China Academy of Railway Sciences Group Co., Ltd under Grant 2021YJ257 and the National Natural Science Foundation of China under Grant 42274189, 42074155 and 42274193.

**Data Availability Statement:** Data available on request due to restrictions e.g., privacy or ethical.

**Acknowledgments:** This work was supported by the Foundation Project of China Academy of Railway Sciences Group Co., Ltd under Grant 2021YJ257 and the National Natural Science Foundation of China under Grant 42274189, 42074155 and 42274193.

**Conflicts of Interest:** The authors declare no conflict of interest.

## Abbreviations

The following abbreviations are used in this manuscript:

| | |
|---|---|
| GPR | Ground Penetrating Radar |
| FDTD | Finite-Difference Time-Domain |
| RIP | Random Irregular Polygons |
| CD | Collision Detection |

## Appendix A. Relative Dielectric Constant Test Method

The relative dielectric constant variation of dry clean ballast to saturation can be obtained in reference [27] as 3–26.9. The approach shown in Figure A1a is used in engineering to determine the relative dielectric constants of the ballast layer, the sand cushion layer, and the subgrade layer. As shown in Figure A1b, which depicts an actual scene for locating the steel plates, numerous steel plates must first be buried to the portions of the medium to be measured, after which the precise buried steel plate depth $D$ is measured. The time parameter $t$ of the depth of the buried steel plate can be determined using GPR, as shown in Figure A1c, and the time required for electromagnetic wave propagation at the buried steel plate can be determined from the radar profile. As a result, we could employ this approach to determine how quickly electromagnetic waves spread through the substance under measurement. We regard the roadbed, sand bedding, and roadbed being measured in this work as homogenous mediums by default in order to make the computation of the relative permittivity easier. Finally, we used the following Equation (A1) to transform the velocity model into a dielectric permittivity model,

$$\varepsilon_r = \left(\frac{c}{v}\right)^2 \tag{A1}$$

where $\varepsilon_r$ is relative dielectric permittivity, $c$ is the speed of light in vacuum and $v = \frac{2D}{t}$ is the subsurface velocity. As a result, the Equation (A1) for the speed at which electromagnetic waves propagate in a material may be used to determine their relative permittivity, as

shown in Table 2. This method allows us to obtain the relative dielectric constants of the ballast layer, the sand cushion layer, and the subgrade layer, which we then include in the simulation.

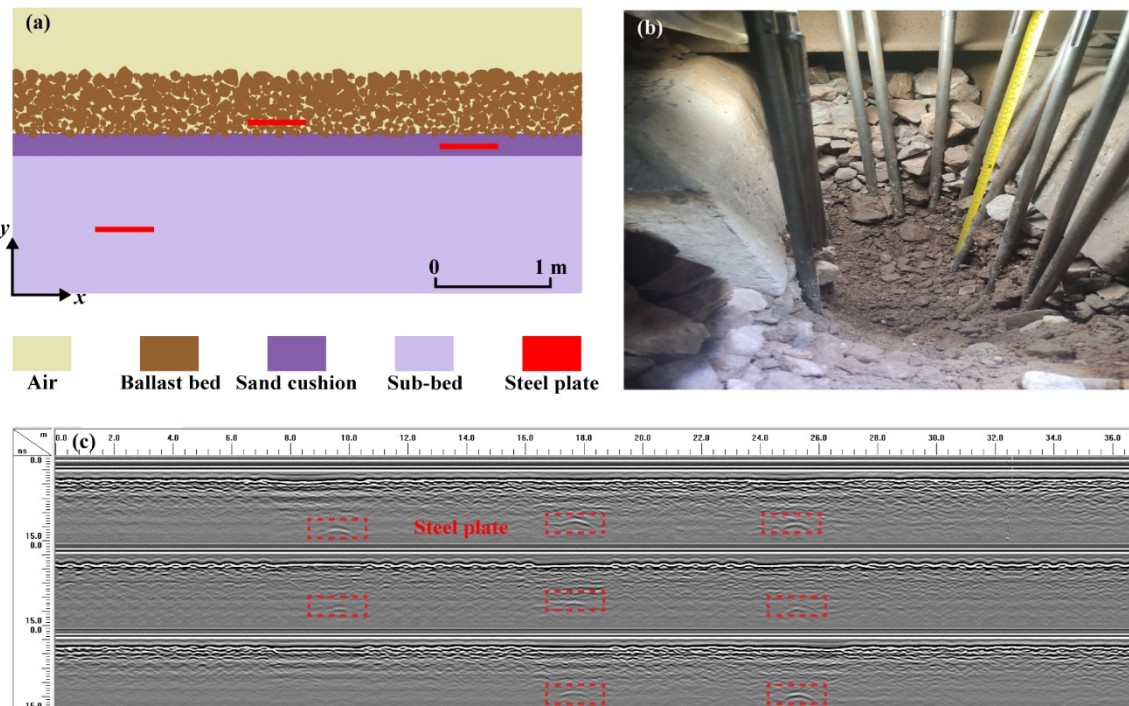

**Figure A1.** (**a**) A Schematic of the relative dielectric constant test, where the red lines all represent steel plates. (**b**) Measurement of ballast bed depth, and (**c**) Three-channel GPR (as shown in Figure 8a) detection results for the actual railroad scene with 8 steel plates buried shown as red dash rectangle.

**Appendix B. Data Processing**

In the study, we processed the simulation data of the ballast bed with different degrees of fouling and mechanical ballast cleaning as well as the measured engineering data of mechanical ballast cleaning. The steps of data processing consist of several steps.

(1) Background elimination. The average value of each segment of the traces was subtracted to complete this stage, which simply included removing the background mean (Figure A2b,e).

(2) Automatic control of gain. Radar echoes steadily lose energy as propagation depth increases. In order to increase the energy of radar echoes from deep reflectors, we routinely utilized automated gain control (Figure A3b).

(3) Railway sleeper interference removal. After the background is removed, the sleeper interference intercepts the position of the data row where the sleeper is located to make it 0 (Figure A2c,f).

(4) Data normalization process. Divide all of the variables to be compared by the highest value of these values to normalize the comparison (Figure 9).

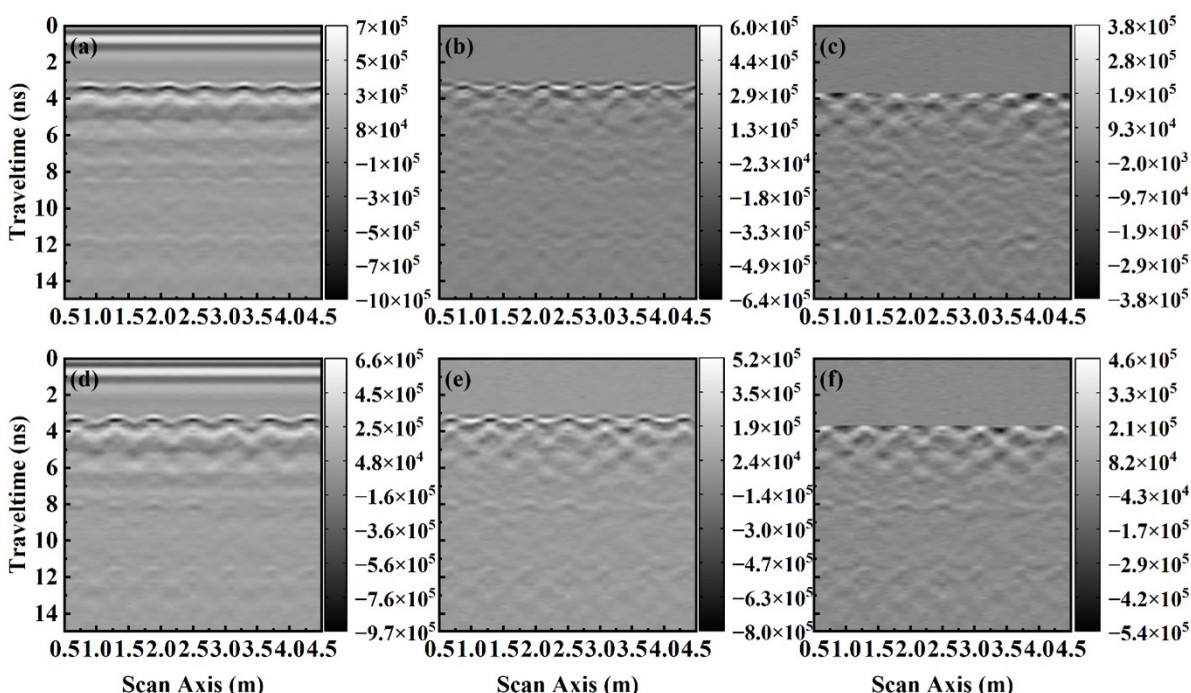

**Figure A2.** The experimental data before and after ballast cleaning radar profiles after applying a series of data regulation processing. (**a**,**d**) The original data. (**b**,**e**) Background elimination. (**c**,**f**) Railway sleeper interference removal.

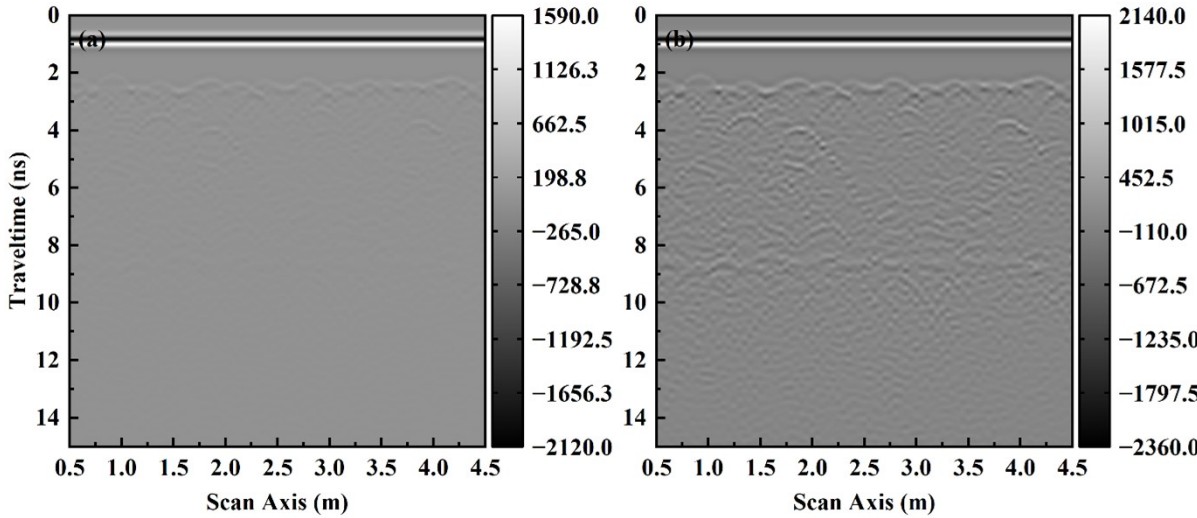

**Figure A3.** Simulation results of clean ballast layers radar profile. (**a**) The original data. (**b**) Automatic control of gain.

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
