# Peer review of "Identification of Ballast Fouling Status and Mechanized Cleaning Efficiency Using FDTD Method"

_remotesensing, doi:10.3390/rs15133437_

Round 1

Reviewer 1 Report

Overview of the paper:

The authors claim that this research paper introduces an innovative approach for modeling railway ballast using random irregular polygons (RIP) and a collision detection (CD) algorithm. The ballast layer is simulated using the finite-difference time-domain (FDTD) algorithm.

Hilbert transform energy, S transform, and energy integration curve results are utilized to determine the extent of ballast fouling and the effectiveness of mechanized cleaning. In the case of highly fouled or untreated ballast layers, the distribution of Hilbert transform energy is more focused, and the S transform exhibits an increased rate of energy attenuation with depth, predominantly concentrated within the 1.0-3.0 GHz range, accompanied by a more pronounced energy integration curve.

On the other hand, in the case of a clean ballast layer or after undergoing a mechanical cleaning process, the results exhibit contrasting patterns. The authors claim that the experiments conducted on a passenger special trunk line in southern China showed favorable agreement between the measured data and simulation results, particularly after the implementation of mechanized ballast cleaning. This method has the potential to serve as a guiding framework for GPR-based detection and offer vital data support for the maintenance of railways.

Although the paper is of certain value in its field and the outcomes are promising, it needs to undergo a minor revision in my view.

1.      The introduction part handled the subject well, but it should be supported by more research. In my opinion there is more research to be referenced regarding the Hilbert transform and time-frequency analyses. If the researchers cannot find them, I can suggest such references. Bottom line, please enrich the list of references.

2.      Line 28-30; The sentence “In railway engineering,……….. to the low resolution.” Ä°s better supported by related references

3.      The first 10 references are cited in appropriate order well-placed throughout the text. However, according to my observation the remaining references (11-20) were either referenced at wrong places or  the topic were not related to the related reference. Therefore I strongly recommend the authors revising this section including those references. Some examples below are (there might be more than the ones below);

·        Line 37; A. Benedetto et al. is cited as reference # 11, but the reference #11 is “Cao, Q.; Al-Qadi, I.L. Effect of Moisture Content on Calculated Dielectric Properties of Asphalt Concrete Pavements from Ground-Penetrating Radar Measurements. Remote Sensing 2021, 14, 34, doi:10.3390/rs14010034.” Maybe in the paper by Cao and Al-Qadi, the issue proposed by A. Benedetto could be referenced and the authors used only Cao and Al-Qadi’s paper, but the original reference should also be mentioned.

·        Same for Line 39; M. Benedetto is mentioned as reference #12 in the main text, but in the reference list reference #12 seems as “ Leshchinsky, B.; Ling, H.I. Numerical Modeling of Behavior of Railway Ballasted Structure with Geocell
Confinement. Geotextiles and Geomembranes 2013, 36, 33–43, doi:10.1016/j.geotexmem.2012.10.006. Maybe in the mentioned paper by Leshchinsky and Ling, the topic proposed by M.Benedetto could be referenced and the authors used only Cao and Al-Qadi’s paper, but the original reference should also be mentioned.

·        The same paper is referecend as Reference #1 and reference # 15. Please correct it.

·        Line 46; Referrence # 16 is actually about Pavement Moisture Content Prediction, but the authors cite it as an exploration research of subsurface water on Mars.

·        Line 85, Please ee-check  the relevance of the text and the references [12-16]

·        The authors should revise such anomalies regarding all the references (particularly the references numbers between and including 11-20)

4.      I recommend authors revelaing the method used to determine the dielectric constants mentioned in Lines 79 and 80.

5.      Regarding Table 1; There are many ballast fouling indices used worldwide. Authors may wish to use them also and at least refer to them while using such a gradation in Table 1 and construction fouling levels. Actually Table 1, is not very clear. It needs better justification with respect to the fouling indices (such as Selig and Water fouling index, Ionescu foulind index, volumetric fouling index, etc.,).

6.      Line 233-242; More specific description and more detailed presentation are required for the processing steps. For example, I recommend authors presenting radargrams before and after each processing steps; especially for truncation (including the other processing steps of course)

7.      Unfortunately, almost no discussion part exists in this manuscript except for a comparison table (Table 3). Please add a robust and justified discussion part with satisfactory commentary on the results.

8.      The authors claim to propose a novel model, but there is literally no mention of the limitations on the capability and the utilizing domain of this model. Please add all the limitations.

9.      The conclusions should include concise extracts of the results and discussions and should be in bulleted style for each conclusion.

10.   Line 267; some sense of caution could be required when asserting that “the method is proposed for the first time”. So please add at the beginning of that sentence “According to the authors ‘best knowledge”.

11.   Line 269 &270 Authors might wish to remove the phrase “great research value” since it is an highly assertive phrase and this should be judged by the readers.

12.   My other concern is the English language of the presented paper. Although, the level of the English language is moderate, a revision of the current state of English made by a native language speaker and/or by a professional who has full proficiency in English, will be appropriate.  Some examples below are (there might be more than the ones below);

·        Line 101-102; The sentence “This process mainly……… and CD.” The part “and CD“ could be removed since it sounds like double mention within the same sentence.

·        Line 124; “be expressed as equation (5)”. Please use “in” instead of “as”.

·        Line 201; please use “changes occurred by mechanized ballast cleaning” instead of “changes mechanized ballast cleaning”.

·        Line 216; Please add “case” to the end of the phrase “before ballast cleaning” such that it would read as “before ballast cleaning case.”

·        Line 217; Please revise the phrase “subjected to mechanized ballast cleaning process efficiency” such that it would read as “subjected to an efficient mechanized ballast cleaning process”.

·        Line 257; Please revise the phrase “judgment of the subsurface GPR for the ballast fouling in practical engineering.” such that it would read as “judgment of the subsurface GPR-based ballast fouling detection in practical engineering.”.

13.   Other minor things

·        Please select a better figure for Figure 1(b). That is not a clear photo representing the caption of that figure.

·        Line 106; It should be Figure 3(b) instead of Figure 2(b)

·        Figure 7 (a) Authors might wish to comment on the frequency range (the abscissa of that graph.

·        Figures 7 (b) and 7 (c), it is not very clear from the text that these figures are for clean or fouled ballast. Please clarify.

·        Line 243, Probably it should be Figure 9 (a), (b), and (c) instead of Figure 9.

·        Line 247, Please revise the sentence for better understanding. Please try simpler expression that reflect the idea related to those figures.

I wrote my comments on this in item 12 above ( in the Comments and Suggestions for Authors part)

Author Response

Responses to Reviewers’ Comments

Q1. The introduction part handled the subject well, but it should be supported by more research. In my opinion there is more research to be referenced regarding the Hilbert transform and time-frequency analyses. If the researchers cannot find them, I can suggest such references. Bottom line, please enrich the list of references.

Author response: 

We appreciate you very much for this important suggestion. In response to your comments, we have expanded the list of references in the manuscript's introduction and included references to the Hilbert energy transform and S-transform time-frequency approaches.

Author action:

We have highlighted the supplementary literature content [19-23] in blue under the references section following your recommendation.

In the re-submitted manuscript Reference Section on page 13.

Lines 362-372,

[19] Guo, Y.; Liu, G.; Jing, G.; Qu, J.; Wang, S.; Qiang, W. Ballast Fouling Inspection and Quantification with Ground Penetrating Radar (GPR). International Journal of Rail Transportation 2023, 11, 151–168, doi:10.1080/23248378.2022.2064346.

[20] Gonzalez, I.; Karoumi, R. Analysis of the Annual Variations in the Dynamic Behavior of a Ballasted Railway Bridge Using Hilbert Transform. Engineering Structures 201460, 126–132, doi:10.1016/j.engstruct.2013.12.026.

[21] Wang, S.; Peng, Z.; Liu, G.; Qiang, W.; Zhang, C. A Study on Characteristic Indexes of Railway Ballast Bed under High-Frequency Radar. Railway Sciences 2023, 2, 33–47, doi:10.1108/RS-02-2023-0009.

[22] Bi, W.; Zhao, Y.; Shen, R.; Li, B.; Hu, S.; Ge, S. Multi-Frequency GPR Data Fusion and Its Application in NDT. NDT & E International 2020, 115, 102289, doi:10.1016/j.ndteint.2020.102289.

[23] Lin, J.; Zhou, W.; Cui, X.Z.; Hong, H.P. Application of Wavelet Transforms to the Simulation of Corrosion Fields on Buried Pipelines. Computers & Structures 2023, 276, 106957, doi:10.1016/j.compstruc.2022.106957.

Q2. Line 28-30; The sentence “In railway engineering,……….. to the low resolution.” Ä°s better supported by related references

Author response: 

We sincerely appreciate your constructive suggestions. We included pertinent references [7] in accordance with your advice to strengthen lines 28-30.

.

Author action:

We have added reference [7] to support lines 28-30.

Lines 28-30,

In the re-submitted manuscript Section 1 on page 2 and Reference Section on page 12

“In railway engineering, early researchers used low-frequency GPR for testing [7], but many results were difficult to interpret due to the low resolution.”

Lines 329-331,

[7] Roberts, R.; Schutz, A.; Al-Qadi, I.; Tutumluer, E. Characterizing Railroad Ballast Using GPR: Recent Experiences in the United States. In Proceedings of the 2007 4th International Workshop on, Advanced Ground Penetrating Radar; June 2007; pp. 295–299.

Q3. The first 10 references are cited in appropriate order well-placed throughout the text. However, according to my observation, the remaining references (11-20) were either referenced at wrong places or the topic were not related to the related reference. Therefore I strongly recommend the authors revising this section including those references. Some examples below are (there might be more than the ones below);

  • Line 37; A. Benedetto et al. is cited as reference # 11, but the reference #11 is “Cao, Q.; Al-Qadi, I.L. Effect of Moisture Content on Calculated Dielectric Properties of Asphalt Concrete Pavements from Ground-Penetrating Radar Measurements. Remote Sensing 2021, 14, 34, doi:10.3390/rs14010034.” Maybe in the paper by Cao and Al-Qadi, the issue proposed by A. Benedetto could be referenced and the authors used only Cao and Al-Qadi’s paper, but the original reference should also be mentioned.
  • Same for Line 39; M. Benedetto is mentioned as reference #12 in the main text, but in the reference list reference #12 seems as “ Leshchinsky, B.; Ling, H.I. Numerical Modeling of Behavior of Railway Ballasted Structure with Geocell Confinement. Geotextiles and Geomembranes 2013, 36, 33–43, doi:10.1016/j.geotexmem.2012.10.006. Maybe in the mentioned paper by Leshchinsky and Ling, the topic proposed by M.Benedetto could be referenced and the authors used only Cao and Al-Qadi’s paper, but the original reference should also be mentioned.
  • The same paper is referecend as Reference #1 and reference # 15. Please correct it.
  • Line 46; Referrence # 16 is actually about Pavement Moisture Content Prediction, but the authors cite it as an exploration research of subsurface water on Mars.
  • Line 85, Please ee-check the relevance of the text and the references [12-16]
  • The authors should revise such anomalies regarding all the references (particularly the references numbers between and including 11-20)

Author response: 

Thank you for your helpful advice. The introduction and references were verified again for consistency, and the amended literature in the re-submitted manuscript was marked in blue.

Author action:

  • Lines 36-38 the reference on A. Benedetto et al. [12] has been revised and replaced.

In the re-submitted manuscript Section 1 on page 2

[12] Benedetto, A.; Tosti, F.; Bianchini Ciampoli, L.; Calvi, A.; Brancadoro, M.G.; Alani, A.M. Railway Ballast Condition Assessment Using Ground-Penetrating Radar – An Experimental, Numerical Simulation and Modelling Development. Construction and Building Materials 2017, 140, 508–520, doi:10.1016/j.conbuildmat.2017.02.110. (Lines 340-343)

  • Line 39, the reference to L. Bianchini Ciampoli et al. [13] has been revised and replaced.

In the re-submitted manuscript Section 1 on page 2

[13] Bianchini Ciampoli, L.; Tosti, F.; Brancadoro, M.G.; D’Amico, F.; Alani, A.M.; Benedetto, A. A Spectral Analysis of Ground-Penetrating Radar Data for the Assessment of the Railway Ballast Geometric Properties. NDT & E International 2017, 90, 39–47, doi:10.1016/j.ndteint.2017.05.005 (Lines 344-346)

  • We remove the duplicate references [15] from the initial manuscript submission.
  • Line 45 (Line 46 in the previous manuscript), corresponding references [17, 18] have been added and replaced.

[17] Wang, Y.; Feng, X.; Liang, W.; Zhou, H.; Dong, Z.; Li, X.; Xue, C. Numerical Simulation of Subsurface Penetrating Radar in Isidis Planitia on Mars for China’s First Mission to Mars. IOP Conf. Ser.: Earth Environ. Sci. 2021, 660, 012024, doi:10.1088/1755-1315/660/1/012024. (Lines 356-358)

[18] Wang, Y.; Feng, X.; Zhou, H.; Dong, Z.; Liang, W.; Xue, C.; Li, X. Water Ice Detection Research in Utopia Planitia Based on Simulation of Mars Rover Full-Polarimetric Subsurface Penetrating Radar. Remote Sensing 2021, 13, 2685, doi:10.3390/rs13142685. (Lines 359-361)

  • Line 92 (Line 85 in the previous manuscript), removed incorrectly labeled reference citations
  • We have carefully checked and revised all references and reordered the references.

Q4. I recommend authors revelaing the method used to determine the dielectric constants mentioned in Lines 79 and 80.

Author response: 

Thank you very much for your advice. Steel plates are buried in the actual ballast bed, sand cushion layer, and subgrade layer in order to define the test technique for the relative permittivity experiment of the ballast layer model. In general, the relative permittivity of the medium may be determined by knowing the steel plate's burial depth and time position in the GPR BScan, as well as the speed at which electromagnetic waves propagate through the medium.

Author action:

In the resubmitted manuscript, it was added that the method regarding the relative permittivity determination is extremely limited. The modified parts are highlighted in blue.

In the re-submitted manuscript on lines 82-87

“The relative dielectric constant test method is mainly through the burial of steel plates in the ballast layer, the sand cushion layer, and the subgrade layer of known depth, using GPR to obtain the propagation time, then the speed of electromagnetic wave propagation in the medium can be obtained, and then according to the formula of electromagnetic wave propagation speed in the medium can be derived from the simulation of the dielectric constant of the ballast layer, the sand cushion layer and the subgrade layer, which based on the mediums are homogeneous and isotropic.”

Q5. Regarding Table 1; There are many ballast fouling indices used worldwide. Authors may wish to use them also and at least refer to them while using such a gradation in Table 1 and construction fouling levels. Actually Table 1, is not very clear. It needs better justification with respect to the fouling indices (such as Selig and Water fouling index, Ionescu foulind index, volumetric fouling index, etc.,).

Author response: 

Thank you very much for your suggestions. The measure of ballast fouling has indeed been extensive. However, Table 1 given in this paper shows the measured ballast particle size results of the railroad bed at different levels of fouling, which helps to directly validate the selected railroad passenger mainline and match the proposed simulation method, which will also mainly serve the proposed algorithm. In addition, the particle size data in this Table 1 are derived from the actual measurement data of selected railroad trunk lines.

Q6. Line 233-242; More specific description and more detailed presentation are required for the processing steps. For example, I recommend authors presenting radargrams before and after each processing steps; especially for truncation (including the other processing steps of course)

Author response: 

Thank you very much for your helpful suggestions. To ensure the authenticity of the data and to retain sufficient data characteristics, the GPR echo signal was first zero-corrected, followed by the deletion of direct waves and the deduction of strong reflected electromagnetic signals from the rail sleepers.

Author action:

Lines 240-243 (Lines 233-242 in the previous manuscript), we add the step for zero-correction.

“For this part of the data, the direct waves of the experimental data were first zero-corrected and removed, and the repeated sleeper information in the experiment was eliminated by truncation, so the results in Figure 9 (a), (b), (d), (e) are relatively flat at around 4 ns.”

Q7. Unfortunately, almost no discussion part exists in this manuscript except for a comparison table (Table 3). Please add a robust and justified discussion part with satisfactory commentary on the results.

Author response: 

Thank you very much for your advice. The relative limitations of both this research and other investigations have been thoroughly analyzed and improved.

Author action:

We additionally highlighted and underlined in blue the relative flaws of this study and related studies.

Line 272-276,

“Although the two-dimensional ballast layer model is straightforward and simple to replicate, there is still a discrepancy between it and the real three-dimensional model. Since every ballast particle in the simulation is similar to a homogeneous, isotropic medium, there is still some experimental room for improvement. The ballast bed model approach also disregards the impact of other elements, such as the amount of water in the ballast, on the level of fouling.”

Q8. The authors claim to propose a novel model, but there is literally no mention of the limitations on the capability and the utilizing domain of this model. Please add all the limitations.

Author response: 

Thank you very much for your kind reminder. We thoroughly evaluated the limitations of this study and included them in the comparison in Table III, as well as stressing this limitation and future research that should be optimized here in the conclusion.

Author action:

Lines 272–276 and Lines 295–301 in the conclusion section now include the flaws of this study and the subsequent research, which we added after careful consideration. The revised manuscript has all changes highlighted in blue.

Lines 272–276

Although the two-dimensional ballast layer model is straightforward and simple to replicate, there is still a discrepancy between it and the real three-dimensional model. Since every ballast particle in the simulation is similar to a homogeneous, isotropic medium, there is still some experimental room for improvement. The ballast bed model approach also disregards the impact of other elements, such as the amount of water in the ballast, on the level of fouling.

Lines 297–303

Nevertheless, the ballast layer model proposed in this paper is still a two-dimensional structure compared to the actual three-dimensional model situation there are still some differences. The relative permittivity of the ballast is only considered as a homogeneous and isotropic medium for the simulation and more environmental effects on the fouling of the ballast bed have not been considered. To address these flaws, further studies will expand the model to a three-dimensional structure and take into account the variation of relative permittivity and some natural climatic effects such as rainfall.

Q9. The conclusions should include concise extracts of the results and discussions and should be in bulleted style for each conclusion.

Author response: 

Thank you very much for your helpful suggestions. To communicate the conclusions and limitations of the study in a more understandable manner, we have improved and amended the conclusions and described the study's contents, methodology, and results in a bulleted style for each conclusion.

Author action:

Lines 278-305

“This paper presents a physically-based railway ballast model for different levels of fouling and mechanized ballast cleaning process efficiency, implemented through a novel Python algorithm using the gprMax simulation software.

  • Based on the proposed method, this study employs the energy integration curve, Hilbert energy transform, and S-transform approaches to evaluate and provide a reasonable analysis of the simulated results on differenct ballast conditions. It can effectively distinguish the railway ballast with different levels of fouling and mechanized ballast cleaning process. These simulation analysis conclusions are consistent with experimental data from a high-speed railway line in southern China, demonstrating the reliability and accuracy of the proposed model.
  • As the fouling increases, the finely fragmented particles in the ballast layer tend to be more abundant and the energy on the integration curve is higher. In the Hilbert transform energy, the energy distribution is more concentrated. In the S transform time-frequency results, the attenuation rate is faster with increasing depth. Conversely, the clean ballast, the opposite effect is observed.
  • This method is proposed can accurately reproduce the railway ballast model with high precision. Furthermore, the proposed research method accurately constructs models of different levels of fouling and mechanized ballast cleaning efficiency of the railway ballast, which has great potential application value in the detection and elimination of hidden dangers on actual railway lines.
  • Nevertheless, the ballast layer model proposed in this paper is still a two-dimensional structure compared to the actual three-dimensional model situation there are still some differences. The relative permittivity of the ballast is only considered as a homogeneous and isotropic medium for the simulation and more environmental effects on the fouling of the ballast bed have not been considered. To address these flaws, further studies will expand the model to a three-dimensional structure and take into account the variation of relative permittivity and some natural climatic effects such as rainfall.}

Finally, the method proposed in this paper could provide more scientifically accurate guidance and explanation for the actual measurement of GPR data on railway lines.

Q10. Line 267; some sense of caution could be required when asserting that “the method is proposed for the first time”. So please add at the beginning of that sentence “According to the authors ‘best knowledge”.

Author response: 

Thank you very much for your kind reminder. We have modified the conclusion section and circumvented this form of description.

Q11. Line 269 &270 Authors might wish to remove the phrase “great research value” since it is an highly assertive phrase and this should be judged by the readers.

Author response: 

Thank you very much for your kind reminder. This type of description has been avoided through modifications to the conclusion section.

Q12. My other concern is the English language of the presented paper. Although, the level of the English language is moderate, a revision of the current state of English made by a native language speaker and/or by a professional who has full proficiency in English, will be appropriate. Some examples below are (there might be more than the ones below);

  • Line 101-102; The sentence “This process mainly……… and CD.” The part “and CD“ could be removed since it sounds like double mention within the same sentence.
  • Line 124; “be expressed as equation (5)”. Please use “in” instead of “as”.
  • Line 201; please use “changes occurred by mechanized ballast cleaning” instead of “changes mechanized ballast cleaning”.
  • Line 216; Please add “case” to the end of the phrase “before ballast cleaning” such that it would read as “before ballast cleaning case.”.
  • Line 217; Please revise the phrase “subjected to mechanized ballast cleaning process efficiency” such that it would read as “subjected to an efficient mechanized ballast cleaning process”.
  • Line 257; Please revise the phrase “judgment of the subsurface GPR for the ballast fouling in practical engineering.” such that it would read as “judgment of the subsurface GPR-based ballast fouling detection in practical engineering.”.

Author response: 

We sincerely appreciate your helpful advice. We invited native English-speaking foreign learners to edit the entire piece in order to enhance our English grammar and skills. In addition, we reviewed the entire content several times for errors to prevent those already noted. All changes are highlighted in blue in the resubmitted manuscript.

Author action:

  • Lines 108-109 (Lines 101-102 in the previous manuscript), we removed the redundant "CD"

“This process mainly involves the CD between ballast particles and the speed of particle distribution.”

  • Line 131 (Line 124 in the previous manuscript), “as” is replaced with “in”.

“can be expressed in equation (5)”

  • Line 207-210 (Line 201 in the previous manuscript), “changes mechanized ballast cleaning” is replaced with “changes occurred by mechanized ballast cleaning”.

“In order to better highlight the contrast effect of energy changes occurred by mechanized ballast cleaning efficiency, this study calculated and obtained the integrated energy of each trace line, plotted the energy curve, and smoothed and normalized the energy values to a range of 0-1, as shown in Figure 7(a).”

  • Lines 222-223 (Line 216 in the previous manuscript), we add the “case” to the phrase.

“By comparing the results, it can be analyzed from the energy decay rate that the 1-6 GHz energy of the after ballast cleaning case generally decays more slowly, while the energy decay rate of the before ballast cleaning case is extremely fast, which is helpful to distinguish and judge whether the railway section has been subjected to an efficient mechanized ballast cleaning process.”

  • Lines 222-224 (Line 217 in the previous manuscript), we have revised the phrase “subjected to mechanized ballast cleaning process efficiency” such that it would read as “subjected to an efficient mechanized ballast cleaning process”.

“By comparing the results, it can be analyzed from the energy decay rate that the 1-6 GHz energy of the after ballast cleaning case generally decays more slowly, while the energy decay rate of the before ballast cleaning case is extremely fast, which is helpful to distinguish and judge whether the railway section has been subjected to an efficient mechanized ballast cleaning process.”

  • Lines 264-265 (Line 247 in the previous manuscript), we have revised the phrase “judgment of the subsurface GPR for the ballast fouling in practical engi-neering.” such that it would read as “judgment of the subsurface GPR-based ballast fouling detection in practical engineering.”

“judgment of the subsurface GPR-based ballast fouling detection in practical engineering.”

Q13. Other minor things

  • Please select a better figure for Figure 1(b). That is not a clear photo representing the caption of that figure.
  • Line 106; It should be Figure 3(b) instead of Figure 2(b)
  • Figure 7 (a) Authors might wish to comment on the frequency range (the abscissa of that graph.
  • Figures 7 (b) and 7 (c), it is not very clear from the text that these figures are for clean or fouled ballast. Please clarify.
  • Line 243, Probably it should be Figure 9 (a), (b), and (c) instead of Figure 9.
  • Line 247, Please revise the sentence for better understanding. Please try simpler expression that reflect the idea related to those figures.

Author response: 

Thank you very much for your kind reminder. We have carefully proofread the entire manuscript regarding the mentioned images and figure notes and made corresponding corrections, which are highlighted in blue.

Author action:

  • We repeatedly exported Figure 1(b) is to tif image format, but the result is not satisfactory and there is still some blurring.
  • Line 113 (Line 106 in the previous manuscript), We have revised Figure 3(b) instead of Figure 2(b).
  • We have revised Figure 7(a), The x-axis label in Figure 7(a) has been changed to "Scan Axis (m)".
  • To clarify Figure 7(b) and (c), we modified Figure 7(a) and then integrated it with the figure caption.
  • Line 242 (Line 243 in the previous manuscript), we revised Figure 9 (a), (b), and (c) instead of Figure 9.

“As shown in Figure 9 (a), (b), and (c), they respectively correspond to the BScan grayscale image, Hilbert transform energy image, and S transform time-frequency image before ballast cleaning.”

  • Lines 253-256 (Line 247 in the previous manuscript), we have revised the sentence for better understanding.

“Before ballast cleaning, the fine particles were primarily found in the lower layer. This resulted in a dense energy distribution in the Hilbert transform shown in Figure 9 (b) and (e). However, after ballast cleaning, the energy distribution became relatively sparse, with the majority concentrated in the surface layer of the ballast.”

Reviewer 2 Report

1.       In abstract, the background and meaning of research are not mentioned. Please figure out in the first to second sentences. Then, please introduce the existing problems in this research.

2.       The main contribution of this paper is suggested to move to the end of introduction.

3.       The experimental conclusion is not summarized.

4.       In the end, the flaw of the paper is not given.

 Moderate editing of English language required

Author Response

Responses to Reviewers’ Comments

Q1. In abstract, the background and meaning of research are not mentioned. Please figure out in the first to second sentences. Then, please introduce the existing problems in this research.

Author response: 

We sincerely appreciate your helpful suggestions. In the first two sentences of the abstract, the background of ground-penetrating radar (GPR) ballast bed inspection and the high cost and traffic obstruction due to ballast bed inspection and excavation tests have been explained. The flaws of this work are that the simulation is only carried out around a two-dimensional roadbed model simulation, and secondly, the relative non-homogeneity of dielectric constants of ballast particles and environmental effects are not fully verified, which will be carried out in the subsequent work.

Author action:

The resubmitted manuscript restates the background and issues of the first two sentences of the abstract. The next phase of work that will be carried out on the deficiencies of this paper is added in the last sentence. The revisions are all highlighted in blue.

Lines 1-4

Systematic assessment of ballast fouling and mechanized cleaning efficiency through ground penetrating radar (GPR) is vital to ensure track stability and safe train transportation. Nevertheless, conventional ballast fouling inspection and evaluation methods impede construction progress and escalate costs.

Lines 12-13

Future studies will consider heterogeneous properties and the three-dimensional structure of the ballast layer.

Q2. The main contribution of this paper is suggested to move to the end of introduction.

Author response:

Thank you very much for your suggestions. We integrate the latter two ends of the introductory group into one paragraph and list the main research contributions of this paper sequentially using a bulleted style.

Lines 50-69,

Inspired by such research, the main contributions of this paper are as follows,}

  • this paper proposes a novel algorithm for generating random irregular polygon (RIP) particles to simulate railway ballast particles, addressing the issue of uniform and unrealistic particle shapes. In addition, an efficient algorithm is presented to generate the ballast layer by collision detection (CD) of a large number of particles, and then the particle size distribution is controlled to simulate the gradation of the ballast. The generated geological models can represent different levels of ballast fouling and the cleaning efficiency of ballast.
  • Using FDTD algorithm in forward simulation, the numerical simulation results accurately determine and identify the differences between the ballast layers with different ballast fouling and mechanized ballast cleaning process efficiency by integrating the energy curve, Hilbert transform energy [19-21], and S-transform time-frequency analysis [22-23]. Finally, we conducted experiments on a section of a high-speed railway line in southern China with screened ballast. By comparing the results of the GPR forward simulation and the experimental data, we found that the simulation results were consistent with the measurements, indicating the accuracy and reliability of the proposed model.
  • The remainder of this paper is organized as follows. Section 2 presents the materials, basic algorithms, and principles of data analysis used in this study. Section 3 provides an analysis of the simulation results for different levels of fouling and before and after ballast cleaning models. Section 4 describes the experimental content designed to compare the simulation results presented in Section 3. Finally, Section 5 summarizes the main contributions of this study

Q3. The experimental conclusion is not summarized.

Author response: 

Thank you very much for your insightful suggestions. The analysis and conclusions about the experiments in this paper are in Section 4. To highlight the consistency of the experiments and simulations, we further generalize the experimental results.

Author action:

Lines 262-265

“Therefore, the processing and analysis of experimental data can fully explain that the ballast model and methods proposed in this study with different ballast fouling and mechanized cleaning processes can interpret well the actual ballast fouling, and guide the analysis and judgment of the subsurface GPR-based ballast fouling detection in practical engineering.”

Q4. In the end, the flaw of the paper is not given.

Author response: 

Thank you very much for your kind reminder. We thoroughly evaluated the limitations of this study and included them in the comparison in Table 3, as well as stressing this limitation and future research that should be optimized here in the conclusion.

Author action:

Lines 270–274 and Lines 295–301 in the conclusion section now include the flaws of this study and the subsequent research, which we added after careful consideration. The revised manuscript has all changes highlighted in blue.

Lines 270–274

“Although the two-dimensional ballast layer model is straightforward and simple to replicate, there is still a discrepancy between it and the real three-dimensional model. Since every ballast particle in the simulation is similar to a homogeneous, isotropic medium, there is still some experimental room for improvement. The ballast bed model approach also disregards the impact of other elements, such as the amount of water in the ballast, on the level of fouling.”

Lines 295–301

“Nevertheless, the ballast layer model proposed in this paper is still a two-dimensional structure compared to the actual three-dimensional model situation there are still some differences. The relative permittivity of the ballast is only considered as a homogeneous and isotropic medium for the simulation and more environmental effects on the fouling of the ballast bed have not been considered. To address these flaws, further studies will expand the model to a three-dimensional structure and take into account the variation of relative permittivity and some natural climatic effects such as rainfall.”

Reviewer 3 Report

In this paper, the authors provide a physically-based railway ballast model for different levels of fouling and mechanized ballast cleaning process efficiency, implemented through a novel Python algorithm using the gprMax simulation software. The simulation results exhibit good consistency with the experimental test data from a high-speed railway line in southern China. The simulation results were analyzed by energy integration curve, Hilbert transform energy, and S transform time-frequency analysis, which effectively distinguish the railway ballast with different levels of fouling and mechanized ballast cleaning process. Furthermore, the proposed research method accurately constructs models of different levels of fouling and mechanized ballast cleaning efficiency of the railway ballast, which has great potential application value in the detection and elimination of hidden dangers on actual railway lines. 

In general, the idea of this paper technically makes sense; and the manuscript is easy to follow. Hence, I may suggest a minor revision for this work. However, the following issues should be addressed:

Detailed comments: 

1-    There is a need to strengthen the Abstract. The author should improve the abstract to include the following in its body: a brief background, brief description of methods and results.

2-    Page 2, line 37: In Introduction Section: “A. Benedetto et al.” this reference is not listed in the references list. Please include it in the references. 

3-    Page 2, line 39: In Introduction Section: “M. Benedetto et al.” this reference is not listed in the references list. Please include it in the references. 

4-    The introduction should briefly indicate the main contributions of this paper in bullets, before the article structure at the end of the introduction section.

5-    Errors in using the singular and plural forms of nouns Errors in the singular and plural forms of nouns.

6-    The references are sufficient, recent and within the field of specialization associated with the title of the paper.

7-    How did the authors set the parameters for the experimental results?

8-    The authors did not discuss the limitations of the proposed approach, and the potential future directions for improvement.

9-    Language in the whole manuscript requires improvements. There are some grammatical issues throughout the paper.  Some grammatical errors and expressions need to be further improved. For example: in page 11, line 258: “To further emphasize the novelty of the proposed method, this study compared it with some relevant research methods...”

Sincerely; Language in the whole manuscript requires improvements. There are some grammatical issues throughout the paper.  Some grammatical errors and expressions need to be further improved.

Author Response

Responses to Reviewers’ Comments

Q1. There is a need to strengthen the Abstract. The author should improve the abstract to include the following in its body: a brief background, brief description of methods and results.

Author response: 

Thank you very much for your insightful suggestions. We have optimized the abstract structure as you requested and added future research that is insufficient for the current work. The modifications have been highlighted in blue.

Author action:

Lines 1-13,

“Systematic assessment of ballast fouling and mechanized cleaning efficiency through ground penetrating radar (GPR) is vital to ensure track stability and safe train transportation. Nevertheless, conventional ballast fouling inspection and evaluation methods impede construction progress and escalate costs. This paper proposes a novel method using random irregular polygons (RIP) and collision detection (CD) algorithms to model the ballast layer. The finite-difference time-domain (FDTD) algorithm simulates the system. Hilbert transform energy, S transform, and energy integration curve are used to identify ballast fouling and cleaning efficiency. Highly fouled layers exhibit concentrated Hilbert transform energy, increased energy attenuation rate in S transform with depth, and concentration in the 1.0-3.0 GHz range, along with a stronger energy integration curve. Clean layers or post-cleaning show opposite results. Experiments on a passenger trunk line in southern China validated the method's accuracy after mechanized ballast cleaning. This approach guides GPR-based detection and supports railway maintenance. Future studies will consider heterogeneous properties and the three-dimensional structure of the ballast layer.”

Q2. Page 2, line 37: In Introduction Section: “A. Benedetto et al.” this reference is not listed in the references list. Please include it in the references.

Author response: 

We sincerely appreciate your kind reminder. In response to your suggestion, we have carefully cross-referenced the full text with the references and made changes to address your suggestions. In addition, similar issues have been revised accordingly and highlighted in blue in the resubmitted manuscript.

Author action:

Lines 36-38, the reference on A. Benedetto et al. [12] has been revised and replaced.

In the re-submitted manuscript Section 1 on page 2

[12] Benedetto, A.; Tosti, F.; Bianchini Ciampoli, L.; Calvi, A.; Brancadoro, M.G.; Alani, A.M. Railway Ballast Condition Assessment Using Ground-Penetrating Radar – An Experimental, Numerical Simulation and Modelling Development. Construction and Building Materials 2017, 140, 508–520, doi:10.1016/j.conbuildmat.2017.02.110. (Lines 340-343)

Q3. Page 2, line 39: In Introduction Section: “M. Benedetto et al.” this reference is not listed in the references list. Please include it in the references.

Author response: 

We sincerely appreciate your kind reminder. We have carefully compared the whole text to the references in response to your proposal, and we've made modifications to take your comments into account. The resubmitted document also includes comparable concerns that have been appropriately corrected and underlined in blue.

Author action:

Line 39, the reference to L. Bianchini Ciampoli et al. [13] has been revised and replaced.

In the re-submitted manuscript Section 1 on page 2

[13] Bianchini Ciampoli, L.; Tosti, F.; Brancadoro, M.G.; D’Amico, F.; Alani, A.M.; Benedetto, A. A Spectral Analysis of Ground-Penetrating Radar Data for the Assessment of the Railway Ballast Geometric Properties. NDT & E International 2017, 90, 39–47, doi:10.1016/j.ndteint.2017.05.005 (Lines 344-346)

Q4. The introduction should briefly indicate the main contributions of this paper in bullets, before the article structure at the end of the introduction section.

Author response: 

Thank you very much for your insightful suggestions. In response to your suggestion, we have used an unordered list in the last paragraph of the introduction to provide a more visual description of the main contributions of the research in this paper. The modifications have been highlighted in blue.

Author action:

Lines 50-69,

Inspired by such research, the main contributions of this paper are as follows,}

  • This paper proposes a novel algorithm for generating random irregular polygon (RIP) particles to simulate railway ballast particles, addressing the issue of uniform and unre-alistic particle shapes. In addition, an efficient algorithm is presented to generate the ballast layer by collision detection (CD) of a large number of particles, and then the par-ticle size distribution is controlled to simulate the gradation of the ballast. The generated geological models can represent different levels of ballast fouling and the cleaning effi-ciency of ballast.
  • Using FDTD algorithm in forward simulation, the numerical simulation results accurate-ly determine and identify the differences between the ballast layers with different ballast fouling and mechanized ballast cleaning process efficiency by integrating the energy curve, Hilbert transform energy [19-21], and S-transform time-frequency analysis [22-23]. Finally, we conducted experiments on a section of a high-speed railway line in southern China with screened ballast. By comparing the results of the GPR forward simulation and the experimental data, we found that the simulation results were con-sistent with the measurements, indicating the accuracy and reliability of the proposed model.
  • The remainder of this paper is organized as follows. Section 2 presents the materials, basic algorithms, and principles of data analysis used in this study. Section 3 provides an analysis of the simulation results for different levels of fouling and before and after ballast cleaning models. Section 4 describes the experimental content designed to com-pare the simulation results presented in Section 3. Finally, Section 5 summarizes the main contributions of this study

Q5. Errors in using the singular and plural forms of nouns Errors in the singular and plural forms of nouns.

Author response: 

We sincerely appreciate your kind reminder. Based on your suggestion, we checked several times for similar errors and invited native English-speaking international students to touch up the English language. The modifications have been highlighted in blue.

Q6. The references are sufficient, recent and within the field of specialization associated with the title of the paper.

Author response: 

Thank you very much for your helpful suggestions. We have carefully checked all references in the article and added and updated them. The modifications have been highlighted in blue.

Author action:

  • Lines 36-38 the reference on A. Benedetto et al. [12] has been revised and replaced.

In the re-submitted manuscript Section 1 on page 2

[12] Benedetto, A.; Tosti, F.; Bianchini Ciampoli, L.; Calvi, A.; Brancadoro, M.G.; Alani, A.M. Railway Ballast Condition Assessment Using Ground-Penetrating Radar – An Experimental, Numerical Simulation and Modelling Development. Construction and Building Materials 2017, 140, 508–520, doi:10.1016/j.conbuildmat.2017.02.110. (Lines 340-343)

  • Line 39, the reference to L. Bianchini Ciampoli et al. [13] has been revised and replaced.

In the re-submitted manuscript Section 1 on page 2

[13] Bianchini Ciampoli, L.; Tosti, F.; Brancadoro, M.G.; D’Amico, F.; Alani, A.M.; Benedetto, A. A Spectral Analysis of Ground-Penetrating Radar Data for the Assessment of the Railway Ballast Geometric Properties. NDT & E International 2017, 90, 39–47, doi:10.1016/j.ndteint.2017.05.005 (Lines 344-346)

  • We remove the duplicate references [15] from the initial manuscript submission.
  • Line 45 (Line 46 in the previous manuscript), corresponding references [17, 18] have been added and replaced.

[17] Wang, Y.; Feng, X.; Liang, W.; Zhou, H.; Dong, Z.; Li, X.; Xue, C. Numerical Simulation of Subsurface Penetrating Radar in Isidis Planitia on Mars for China’s First Mission to Mars. IOP Conf. Ser.: Earth Environ. Sci. 2021, 660, 012024, doi:10.1088/1755-1315/660/1/012024. (Lines 356-358)

[18] Wang, Y.; Feng, X.; Zhou, H.; Dong, Z.; Liang, W.; Xue, C.; Li, X. Water Ice Detection Research in Utopia Planitia Based on Simulation of Mars Rover Full-Polarimetric Subsurface Penetrating Radar. Remote Sensing 2021, 13, 2685, doi:10.3390/rs13142685. (Lines 359-361)

  • Line 92 (Line 85 in the previous manuscript), removed incorrectly labeled reference citations
  • We have carefully checked and revised all references and reordered the references.
  • Line 85, we removed incorrectly labeled reference citations

We have carefully checked and revised all references and reordered the references. (In re-submitted manuscript Reference Section on page 12-14.)

Q7. How did the authors set the parameters for the experimental results?

Author response: 

Thank you very much for your helpful suggestions. All the experimental results parameters in this study are set based on engineering experience, including the selection of the GPR antenna at 2.0 GHz and the corresponding settings of time window and spacing. You can find more information about the experimental parameter settings in Section 2, lines 75-76 of the article.

Author action:

Q8. The authors did not discuss the limitations of the proposed approach and the potential future directions for improvement.

Author response: 

Thank you very much for your kind reminder. We thoroughly evaluated the limitations of this study and included them in the comparison in Table III, as well as stressing this limitation and future research that should be optimized here in the conclusion.

Author action:

Lines 272–276 and Lines 297–303 in the conclusion section now include the flaws of this study and the subsequent research, which we added after careful consideration. The revised manuscript has all changes highlighted in blue.

Lines 272–276

“ Although the two-dimensional ballast layer model is straightforward and simple to replicate, there is still a discrepancy between it and the real three-dimensional model. Since every ballast particle in the simulation is similar to a homogeneous, isotropic medium, there is still some experimental room for improvement. The ballast bed model approach also disregards the impact of other elements, such as the amount of water in the ballast, on the level of fouling.”

Lines 297–303

“ Nevertheless, the ballast layer model proposed in this paper is still a two-dimensional structure compared to the actual three-dimensional model situation there are still some differences. The relative permittivity of the ballast is only considered as a homogeneous and isotropic medium for the simulation and more environmental effects on the fouling of the ballast bed have not been considered. To address these flaws, further studies will expand the model to a three-dimensional structure and take into account the variation of relative permittivity and some natural climatic effects such as rainfall.”

Q9. Language in the whole manuscript requires improvements. There are some grammatical issues throughout the paper. Some grammatical errors and expressions need to be further improved. For example: in page 11, line 258: “To further emphasize the novelty of the proposed method, this study compared it with some relevant research methods...”

Author response: 

We sincerely appreciate your helpful advice. We invited native English-speaking foreign learners to edit the entire piece in order to enhance our English grammar and skills. In addition, we reviewed the entire content several times for errors to prevent those already noted. All changes are highlighted in blue in the resubmitted manuscript.

Author action:

  • Lines 108-109 (Lines 101-102 in the previous manuscript), we removed the redundant "CD"

“This process mainly involves the CD between ballast particles and the speed of particle distribution.”

  • Line 131 (Line 124 in the previous manuscript), “as” is replaced with “in”.

“can be expressed in equation (5)”

  • Line 207-210 (Line 201 in the previous manuscript), “changes mechanized ballast cleaning” is replaced with “changes occurred by mechanized ballast cleaning”.

“In order to better highlight the contrast effect of energy changes occurred by mechanized ballast cleaning efficiency, this study calculated and obtained the integrated energy of each trace line, plotted the energy curve, and smoothed and normalized the energy values to a range of 0-1, as shown in Figure 7(a).”

  • Lines 222-223 (Line 216 in the previous manuscript), we add the “case” to the phrase.

“By comparing the results, it can be analyzed from the energy decay rate that the 1-6 GHz energy of the after ballast cleaning case generally decays more slowly, while the energy decay rate of the before ballast cleaning case is extremely fast, which is helpful to distinguish and judge whether the railway section has been subjected to an efficient mechanized ballast cleaning process.”

  • Lines 222-224 (Line 217 in the previous manuscript), we have revised the phrase “subjected to mechanized ballast cleaning process efficiency” such that it would read as “subjected to an efficient mechanized ballast cleaning process”.

“By comparing the results, it can be analyzed from the energy decay rate that the 1-6 GHz energy of the after ballast cleaning case generally decays more slowly, while the energy decay rate of the before ballast cleaning case is extremely fast, which is helpful to distinguish and judge whether the railway section has been subjected to an efficient mechanized ballast cleaning process.”

  • Lines 264-265 (Line 247 in the previous manuscript), we have revised the phrase “judgment of the subsurface GPR for the ballast fouling in practical engi-neering.” such that it would read as “judgment of the subsurface GPR-based ballast fouling detection in practical engineering.”

“judgment of the subsurface GPR-based ballast fouling detection in practical engineering.”
